# Binary Dye Removal from Simulated Wastewater Using Reduced Graphene Oxide Loaded with Fe-Cu Bimetallic Nanocomposites Combined with an Artificial Neural Network

**DOI:** 10.3390/ma14185268

**Published:** 2021-09-13

**Authors:** Ling Xin, Xianliang Wu, Yiqiu Xiang, Shengsheng Zhang, Xianfei Huang, Huijuan Liu

**Affiliations:** 1Guizhou Provincial Key Laboratory for Information Systems of Mountainous Areas and Protection of Ecological Environment, Guizhou Normal University, Guiyang 550001, China; xinling901231@163.com (L.X.); xxiangyiqiu@163.com (Y.X.); zss17885509214@163.com (S.Z.); 2Guizhou Institute of Biology, Guiyang 550009, China; 16010160437@gznu.edu.cn; 3Institute for Forest Resources and Environment of Guizhou, College of Forestry, Guizhou University, Guiyang 550025, China; 4The Key Laboratory of Environmental Pollution Monitoring and Disease Control, Guizhou Medical University, Ministry of Education, Guiyang 550025, China

**Keywords:** adsorption, nanocomposite, binary dyes, artificial intelligence, kinetics

## Abstract

Reduced graphene oxide loaded with an iron-copper nanocomposite was prepared in this study, using graphene oxide as a carrier and ferrous sulfate, copper chloride and sodium borohydride as raw materials. The obtained material was prepared for eliminating hazardous dye carmine and the binary dye mixture of carmine and Congo red. The process of carmine dye removal by the nanocomposite was modeled and optimized through response surface methodology and artificial intelligence (artificial neural network–particle swarm optimization and artificial neural network–genetic algorithm) based on single-factor experiments. The results demonstrated that the surface area of the nanocomposite was 41.255 m^2^/g, the pore size distribution was centered at 2.125 nm, and the saturation magnetization was up to 108.33 emu/g. A comparison of the material before and after the reaction showed that the material could theoretically be reused three times. The absolute error between the predicted and experimental values derived by using artificial neural network–particle swarm optimization was the smallest, indicating that this model was suitable to remove carmine from simulated wastewater. The dose factor was the key factor in the adsorption process. This process could be described with the pseudo-second-order kinetic model, and the maximum adsorption capacity was 1848.96 mg/g. The removal rate of the mixed dyes reached 96.85% under the optimal conditions (the dosage of rGO/Fe/Cu was 20 mg, the pH was equal to 4, the initial concentration of the mixed dyes was 500 mg/L, and the reaction time was 14 min), reflecting the excellent adsorption capability of the material.

## 1. Introduction

In recent years, environmental problems arising from dye wastewater pollution have become increasingly prominent [1]. Most dyestuffs with complex aromatic ring structures have toxicity and bioaccumulation, which are difficult to degrade under natural conditions. Among the various dyes, direct dyes are widely used because they can be applied directly without mordant. Congo red, a typical benzidine direct azo dye, easily enters the water due to the high loss rate in the process of production and their application, which has a greatly harmful effect on the environment [2]. Traditional biochemical treatment methods (such as activated sludge) can be used for wastewater treatment, but the effect is not very satisfactory. Carmine is currently the most widely available and largest amount of a single-azo synthetic dye. Recently, it has been reported that carmine and the EU standard banned Sudan red I with azo pigments, since azo compounds in the body can be metabolized to generate mutagenic precursors, namely aromatic amines [3]. Due to the high chromaticity, low biodegradability and high toxicity of dye wastewater, dye wastewater pollution is considered to be one of the challenges facing mankind [4]. The common treatment technologies of dye wastewater include adsorption, ion exchange, coagulation, advanced oxidation, electrochemical and membrane separation technologies. Among these techniques, adsorption is an effective method for treating wastewater containing dyes, with the advantages of simplicity, a relatively low cost and easy operation [5]. It is particularly important to prepare environmentally friendly and efficient adsorbents.

Carbonaceous material adsorbents have broad application prospects in the treatment of dye-containing wastewater [6]. Geim et al. [7] found that new materials have been extensively studied in terms of their properties and applications, especially the unique two-dimensional structure of graphene. Since graphene is a two-dimensional honeycomb planar nanomaterial composed of six-membered rings with only one atomic thickness (0.334 nm), it has a large specific surface area with a theoretical value of up to 2630 m^2^/g. Meanwhile, it has a wide range of applications in the removal of organic and inorganic pollutants from wastewater by adsorption [8,9]. However, graphene generally undergoes stacking and agglomeration, resulting in a large loss of effective surface area and poorer than expected adsorption of the composites, which limits their practical application to some extent. The metal oxides uniformly distributed on the graphene sheets in the binary metal graphene oxide (GO) composites can prevent the stacking and agglomeration of graphene. Meanwhile, graphene can ameliorate the electrical conductivity and increase the dispersion of the binary transition metal oxides. Kang et al. [10] loaded iron ions (II, III) on GO to enhance the activation of peroxynitrite (PS) for the degradation of phenol. The results indicated that PS/GO-Fe (III) was more efficient than PS/GO-Fe (II) based on the comparison of phenol degradation. Hou et al. [11] investigated the decolorization of toxic bright green dye in the aqueous phase by reduced graphene oxide loaded with mesoporous Pd-Fe bimetallic magnetic nanoparticles, and their results indicated that the material was an effective mesoporous material for the purification of wastewater containing a high concentration of dyes. Farooq et al. [12] selected tricarboxylic acids as model chlorinated organic compounds to examine the catalytic activity of mono- and bimetallic (Fe and Fe-Cu) heterogeneous nanocomposites supported by reduced graphene oxide. The influence of the synthesis methodology on the properties and activity of the heterogeneous composites was also investigated, and its excessive effect in improving Fenton-based AOPs was determined. The above-mentioned reports proved that the GO combined with the other metals could overcome its limitations. Therefore, in this research, the reduced graphene oxide loaded with the Fe-Cu bimetallic nanocomposites was prepared for the adsorption of single dyes (carmine) as well as bimetallic dyes (carmine and Congo red) from simulated industrial wastewater.

Artificial intelligence is the research of how to make computers mimic thinking activities such as understanding, learning, reflecting and programming [13]. There are many algorithms for artificial intelligence, such as genetic algorithms (GAs), random forests (RFs), ant colony algorithms (ACAs), augmented regression trees (BRTs), simulated annealing (SA), Monte Carlo simulation (MCS), particle swarm optimization (PSO) and artificial neural networks (ANNs) [14]. Functional approximation theory considers that an approximate function can be used to approximate an unknown actual function. If the series of errors generated in the prediction is also taken as the value of the approximation function, then it is reasonable to assume that the prediction error can be further reduced by re-approximation to improve the validity of the prediction. Hence, a combined neural network model is proposed in an attempt to further reduce the errors arising from the prediction. In the last decade, the research work on artificial neural networks has made great progress, which has successfully addressed many practical problems and demonstrated excellent, intelligent properties. Moradi et al. [15] modeled and optimized the NH_4_^+^ adsorption process by varying four independent parameters (pumice dose, initial ammonium ion concentration, mixing rate and contact time) to get the most optimum conditions by performing a central composite design (CCD) under the response surface methodology (RSM). Kakhki et al. [16] fabricated a sulfur–nitrogen co-doped Fe_2_O_3_ nanostructure to degrade dimethyl blue and evaluated and optimized the experimental conditions for the dose of nanoparticles, concentration of the dye, pH and dose of light. An ANN model was used to predict the removal efficiency of the dyes, where R^2^ was about 92%, so the proposed ANN-GA model achieved acceptable performance. This indicated that artificial neural networks were utilized in many fields, which proved that it was a reliable technique. Our research will combine artificial neural networks with adsorption experiments to predict the optimal conditions as well as the optimal values of the experimental process.

The above research indicates that the removal of pollutants from the water environment can be greatly improved with the assistance of artificial intelligence technology. Therefore, this experiment aimed to (1) remove single (carmine) and dual (carmine and Congo red) dyes with the mesoporous nanomaterial rGO/Fe/Cu; (2) show these nanohybrids were prepared by the co-precipitation method and characterized by X-ray diffraction (XRD), scanning electron microscopy (SEM), transmission electron microscopy (TEM), Fourier Transform Infrared (FT-IR) energy dispersive spectroscopy (EDS), Raman spectroscopy, N_2_ adsorption and X-ray photoelectron spectroscopy (XPS); (3) demonstrate that the RSM was determined by single-factor experiments (reaction time, pH, temperature and initial concentration) to derive experimental conditions, which were then optimized, followed by the AI models (ANN-PSO, ANN-GO, etc.) being used to explore the modeling and optimization of the dye removal process with nanomaterials, verify the removal rate under optimal conditions and rank the importance of the factors; and (4) evaluate the kinetic properties of the dye removal process. The material adsorbed the industrial wastewater simulated by the dye-containing wastewater, improved the drainage quality of wastewater treatment systems and reduced environmental pollution.

## 2. Experiment

### 2.1. Materials

In the preparation process, the sulfuric acid (H_2_SO_4_) and hydrochloric acid (HCl) were of superior purity, while the other reagents were of analytical purity. The H_2_SO_4_, HCl and graphite powder (particle size < 30 µm, purity > 99.85%) were all purchased from Sinopharm Holdings Chemical Reagent Co. (Beijing, China). The potassium permanganate (KMnO_4_) was manufactured by Chongqing Jiangchuan Chemical Co.(Chongqing, China) and the sodium nitrate (NaNO_3_) was produced by Beijing Chemical Factory (Beijing, China). Ferrous sulfate heptahydrate (FeSO_4_·7H_2_O) was purchased from the Chengdu Jinshan Chemical Reagent Co. (Chengdu, China), and copper chloride (CuCl_2_·2H_2_O) was supplied by the Tianjin Ruijinte Chemicals Co. (Tianjin, China) Carmine, Congo red and sodium borohydride (NaBH_4_) were generated by the Tianjin Comio Chemical Reagent Co., (Tianjin, China).

### 2.2. Synthesis of Nanoparticles

#### 2.2.1. Preparation of the GO

Graphene oxide was synthesized by the modified Hummers method [17]. The first stage was the low-temperature period, in which ice packs were added into the water bath to lower its temperature, reaching about 15 °C. Then, 46 mL H_2_SO_4_, 2.0 g of graphite powder, 1.0 g NaNO_3_, and 6.0 g KMnO_4_ were added into a 500-mL beaker at 15 °C in an ice bath under vigorous stirring for 2 h. The second phase was the medium-temperature stage, in which the reaction solution was warmed to 35 °C and stirred for 30 min. The third stage was the high-temperature stage, taking place after the reaction solution temperature rose to 98 °C. Here, 90 mL of deionized water was slowly added under stirring to carry out the reaction for 15 min. Then, 200 mL of deionized water was added to dilute the suspension. Finally, an appropriate amount of H_2_O_2_ (30 wt%) was added into the solution until the suspension turned bright yellow (at 98 °C). The suspension was washed three times with 5% HCl and deionized water (at room temperature) and then centrifuged (4000 rpm, 8 min). It was then dried in a vacuum oven (60 °C, 48 h), ground and set aside [18,19].

#### 2.2.2. Synthesis of rGO/Fe/Cu Nanocomposites

The rGO/Fe/Cu nanohybrids were prepared by the co-precipitation method [20], in which 100 mL of the aqueous solution containing FeSO_4_·7H_2_O (5 g/50 mL) and CuCl_2_·2H_2_O (0.134 g/50 mL) was added into the GO (0.5 g/150 mL) solution and sonicated for 2 h. The mixture was then stirred for 12 h. Next, the NaBH_4_ (2.7 g/25 mL) was added and agitated for 30 min under a nitrogen atmosphere at room temperature. Then, the black precipitate was obtained by vacuum pumping and washed three times with ethanol and deionized water. Finally, the precipitates were dried in a vacuum oven at 60 °C for 24 h, and rGO/Fe/Cu with a molar ratio of 5:1 was successfully manufactured. The Fe/Cu bimetallic compound was fabricated in the same way without the incorporation of GO (Figure 1).

#### 2.2.3. Preparation of Dye Samples

The carmine solution and the mixture of carmine and Congo red were prepared with deionized water. The solutions required for the experiment were diluted by deionized water to the needed initial concentration, and 0.5 g of carmine was added into 500 mL of deionized water to prepare a single dye solution (1000 mg/L). After that, 0.5 g of carmine and 0.5 g of Congo red were added to 500 mL of deionized water to prepare a binary dye solution (2000 mg/L). The structure and properties of the carmine (molecular formula: C_20_H_11_O_10_N_2_S_3_Na_3_, molecular weight: 604, λmax = 489 nm) and Congo red (molecular formula: C_32_H_22_N_6_Na_2_O_6_S_2_, molecular weight: 696.68, λmax = 454 nm) are shown in Figure 2 and Table 1. The maximum absorption wavelength of the mixture of carmine and Congo red was measured at 503 nm under a UV spectrophotometer.

### 2.3. Batch Decontamination Experiments

The effects of the initial dye concentration, pH, reaction time and dose on the decontamination efficiency of single-dye and dual-dye simulated wastewater were investigated by single-factor experiments. Then, the single-dye training experiments, prediction experiments and validation experiments were based on RSM as well as ANN-GA and ANN-PSO. The batch decontamination experiments were carried out in 150-mL conical flasks with the reaction solution volume kept at 50 mL. The stock solutions were diluted to 100–1000 mg/L using deionized water and a reaction time ranging from 2 to 30 min. rGO/Fe/Cu nanocomposites (10–30 mg) were added into 50 mL of the dye solutions (100–1000 mg/L) with an initial pH (3–10). The initial solution pH adjustments were performed with HCl (0.1 mol L^−1^) and NaOH (0.1 mol L^−1^) solutions. The conical flasks were placed into an ultrasonic cleaner (power: 180 W) to sonicate for a certain period. Finally, the rGO/Fe/Cu nanocomposites were separated from the reaction solutions by magnets. The final concentrations of the single-dye and binary-dye solutions were measured using a UV-visible spectrophotometer at a λ_max_ of 509 nm and 503 nm, respectively. The percentage of dye decontamination and the decontamination quantity at equilibrium were calculated by the following equations:(1)$$\;\;R\left(\%\right)=\frac{C_{0}-C_{t}}{C_{0}}*100\%$$
(2)$$q_{e}=\frac{\left(C_{0}-C_{t}\right)*v}{m}$$
where *R* is the decontamination percentage of the dye; *C*_0_ (mg/L) and *C_t_* (mg/L) represent the initial dye concentration and the final dye concentration after decontamination, respectively; *v* is the volume of the solution (mL) and *m* is the dosage of rGO/Fe/Cu nanocomposites (mg).

### 2.4. Characterization

The diffractograms of the prepared Fe/Cu nanoparticles and rGO/Fe/Cu nanocomposites were obtained by an X-ray diffractometer (Lelyweg 1, Almelo, The Netherlands) which was operated with a Cu-K*α* source, a tube voltage of 40 kV, a tube current of 40 mA, a scanning speed of 10°/min and a scanning angle 2θ range of 5–80°. The surface and internal morphological characteristics of the materials were investigated in this research based on the Quanta F250 field emission scanning electron microscope and Tecnai G220 transmission electron microscope manufactured by FEI, Florida, FL, USA. The rGO/Fe/Cu nanocomposites and Fe/Cu nanoparticles were characterized with a Vector 33 infrared spectrometer, Bruker, Germany. The samples were prepared by the KBr press method with a resolution of 4 cm^−1^ and a scan range of 4000–400 cm^−l^. The positions and intensities of the absorption bands in the infrared spectra can reflect the characteristics of the molecular structures. The N_2_ adsorption and desorption isotherms at 77 K (Quantum Star Instruments, Poynton Beach, FL, USA) were used to confirm the surface areas and the narrow pore size distributions of these materials. Magnetization measurements were performed with a squid magnetometer (MPMSXL-7, Quantum Design, Inc., San Diego, CA, USA) at room temperature under an applied magnetic field. The Fe/Cu nanoparticles and rGO/Fe/Cu nanocompounds were observed by X-ray photoelectron spectroscopy with an ESCALAB 250Xi spectrometer (Thermo Electron Corporation, Waltham, MA, USA). 

### 2.5. Response Surface Methodology

In this study, the experimental conditions for decolorization of mesoporous rGO/Fe/Cu magnetic nanocomposites in an aqueous solution were optimized utilizing the Box–Behnken design (BBD) technique under RSM. RSM is a scientific and quantitative analysis method to investigate the relationship between response variables and input variables [21,22]. Through a suitable experimental design scheme, the actual response values of a certain number of points near a point were obtained, and a complex simulation model (i.e., response surface model) was established with a simple functional relationship approximating the actual replacement. The accuracy and efficiency of fitting this approximate model instead of the actual function in a region sufficiently close to this point to perform complex calculations would directly affect the subsequent optimization results. The functional expressions for the first-order and second-order polynomial response surface approximation models are shown in the following forms [23]:(3)$$y=\beta_{0}+\sum_{i=1}^{n}\beta_{i}x_{i}+\varepsilon$$
(4)$$y=\beta_{0}+\sum_{i=1}^{n}\beta_{i}x_{i}+\sum_{i=1}^{n}\beta_{ii}{x_{i}}^{2}+\sum_{i=1}^{n}\sum_{j>i}^{n}\beta_{ij}x_{i}x_{j}\;+\;\varepsilon$$
where *y* stands for the value for the predicted response; *β*_0_ stands for the constant term; *β*_*i*_ denotes the primary term coefficient; *β**_ii_* refers to the quadratic term coefficient; *β**_ij_* serves as the interaction term coefficient; *x_i_* and *x_j_* represent the design variables and *ε* indicates the error term.

The design factors and their level coding for the three-level experimental design of the four design variables (initial concentration, initial pH, reaction time and dose) performed in Design-Expert software and employing the BBD are shown in Table 2, and 29 sets of experimental conditions were obtained, according to Table 2.

### 2.6. Modeling and Optimization of the AI

An ANN is a mathematical model that simulates the processing mechanism of the human brain’s nervous system for complex information based on the basic principle of a biological central neural network and the theoretical basis of network topology knowledge, and the data are fitted by training values, experimental values and predicted values [24,25]. It is a complex network with a large number of simple nodes, both interconnected and transmitted, capable of performing complex logical operations, and it has the four basic characteristics of being highly nonlinear, non-constrained, non-qualitative and nonconvex. In this work, a backpropagation (BP) algorithm was implemented to construct a predictive mathematical model with four factors (i.e., dye concentration, pH, reaction time and dose) [11]. The network had three layers—an input layer, a hidden layer and an output layer—with a tangential S-shaped transfer function for the hidden input layer and a linear transfer function for the hidden output layer (Figure 3).

PSO is a randomized search algorithm that explores the optimal region in space through the accumulation of the individual particles’ own experience and the learning of the group’s excellent information [26,27]. The PSO algorithm regards the control variables as properties of themselves, and it can be efficiently transformed for equation constraints, while inequality constraints can be attached to the objective function in the form of penalty functions. In this article, the experimental data were optimized and predicted utilizing ANN-PSO, and then the predicted conditions were verified. A radial basis network (RBF) belongs to the multilayer forward neural network, which is a three-layer forward network [28,29]. In this task, we evaluated the four main parameters as input data, including reaction time, pH, dosage as well as dye concentration, and then the percentage of carmine removal with an RBF as output data. The RBF was expressed by the Gaussian function [19]:(5)$$\varphi_{ij}=\exp\left(\frac{\|x_{j}-c_{i}{\|}^{2}}{\sigma_{j}^{2}}\right)$$
where *x_j_* represents the input vector, ‖ ‖ is a measure of Euclidean distance and *c_i_* and *σ_j_* are the centers and the spread of the *j*th RBF, respectively.

## 3. Results and Discussion 

### 3.1. Characterization of the rGO/Fe/Cu Nanocompounds

X-ray diffraction (XRD) is an important technique extensively used for characterizing the crystal structure, chemical composition and physical properties of materials [30]. According to Qi et al. [17], the diffraction peak of the GO was around 10°, but this peak disappeared in the compound, indicating that the GO was reduced to reduced graphene oxide (Figure 4a). XRD revealed that the main component was Fe^0^, with a diffraction peak observed at 2θ of 44.6°, and also contained *α*-Fe_3_O_4_ components, with diffraction peaks at 65.0° and 35.5° and a characteristic peak at 2θ of 23.8° CuO [31,32]. Figure 4 shows that there was no excess impurity peak on the surface of the prepared rGO/Fe/Cu, which indicated that the purity of the rGO/Fe/Cu prepared by this method was high. Although the nZVI part of the material surface was oxidized, nZVI and CuO were still dominant.

As demonstrated in Figure 5b, the Fe/Cu nanoparticles were loaded onto a reduced graphene oxide (rGO) sheet. Compared with the Fe/Cu nanoparticles, the dispersion of rGO/Fe/Cu nanohybrids was more uniform and predominant, but the agglomeration still existed. As shown in Figure 5c, the rGO/Fe/Cu nanohybrids could be recovered after the test, and the Fe/Cu nanoparticles were compactly stacked together with only a few areas for the intercalation of dye heterocysts, which indicated that the material was unchanged before and after the reaction and provided the possibility to repeat the experiment. Figure 6 presents the TEM image of rGO/Fe/Cu, which reveals that the prepared rGO/Fe/Cu on the nano-iron was nearly spherical, with a particle size of 20–100 nm, and had an Fe-Fe_3_O_4_ core shell structure similar to that of the nano-iron. However, the prepared rGO/Fe/Cu had significantly better dispersion, a higher specific surface area, more reaction sites and higher reactivity than Fe/Cu. The experiments were conducted by energy-dispersive X-ray spectroscopy (EDS) to further characterize the elemental composition of the rGO/Fe/Cu composites. Elements such as Fe and Cu detected by EDS on the iron surface are displayed in Figure 7, and the weight ratio of the Cu/Fe loaded on the GO was 8.08/40.72, which was consistent with the experimental design value. Nevertheless, the Fe/Cu weight ratio of the unloaded GO differed significantly from the experimental value, and the data of the loaded GO demonstrated that the new rGO/Fe/Cu bimetallic nanocomposites had been successfully fabricated.

In order to investigate the molecular interaction of the dyes with the rGO/Fe/Cu nanohybrids, the FTIR spectra of the rGO/Fe/Cu nanohybrids before and after the dye removal experiment is shown in Figure 8. Several characteristic peaks were observed in the rGO/Fe/Cu spectrum, such as 3436.99 cm^−1^, 1058.08 cm^−1^ and 1633.31 cm^−1^, corresponding to the hydroxyl peaks (-OH), epoxy groups (C-O-C) and bending vibrational peaks of C-O in the C-OH functional groups, respectively, which showed that the rGO surface contained a large number of oxygen-containing functional groups. The absorption peak at 1633.31 cm^−1^ was the C-C stretching vibration peak, and the characteristic peak at 581.03 cm^−1^ was the Fe-O stretching vibration peak, indicating that the bonding of rGO with Fe^0^ nanoparticles was mainly accomplished by Fe-O bonding. The comparison results of the characteristic peaks of the rGO/Fe/Cu nanocompounds before and after the dye removal experiment indicated that the peaks corresponding to the individual functional groups did not disappear but only partially weakened in intensity, which further illustrated that the substance was theoretically reusable with a possible partial reduction in adsorption efficiency. 

The N_2_ adsorption-desorption isotherms of the Fe/Cu nanoparticles and rGO/Fe/Cu nanocompounds are shown in Figure 9, in which the curves of the Fe/Cu nanoparticles and rGO/Fe/Cu nanohybrids belonged to the IV isotherm, which was typical for the mesoporous materials. The adsorption-desorption isotherm of the rGO/Fe/Cu nanocompounds displayed a loop at a relative pressure between 0.2 and 1.0 because of the condensation of nitrogen inside the mesopores. The BJH pore size distribution of the Fe/Cu nanoparticles and rGO/Fe/Cu nanohybrids was centered at 3.930 nm and 3.933 nm, respectively, which revealed that both materials were mesoporous (Figure 10). Both the Fe/Cu nanoparticles and rGO/Fe/Cu nanohybrids had large surface areas of 64.811 and 41.255 m^2^/g, respectively.

Raman spectroscopy was used to prove the physical properties and molecular structure of the rGO/Fe/Cu nanohybrids (Figure 11). The former peak and latter peak were called the D band and G band, respectively. The intensity ratio of the D band to the G band (ID/IG) is commonly referred to as a measure of the level of defects in graphene-based materials, and if ID/IG > 1, there are more structural defects [33]. In this work, the ID/IG intensity (1.18) of the rGO/Fe/Cu nanohybrids was greater than 1, indicating a large number of structural defects in the rGO/Fe/Cu nanohybrids. Figure 11 indicates that the numerous structural defects could provide more adsorption sites for dye adsorption to improve the adsorption efficiency. Figure 12 presented the magnetization properties of the Fe/Co nanoparticles and rGO/Fe/Co nanohybrids. The saturation magnetization intensity of the Fe/Cu nanoparticles and rGO/Fe/Cu nanohybrids were 125.43 emu/g and 108.33 emu/g, respectively. The rGO/Fe/Cu nanohybrids could be quickly recovered from the dye solutions due to their high saturation magnetization. 

X-ray photoelectron spectroscopy (XPS) was employed to investigate the elemental composition and chemical valence states of the ball-ground samples [34]. The binding energies of all spectral peaks were calibrated regarding C1S at a binding energy of 284.6 eV. Figure 13 shows that the binding energy of the Cu 2p peak was 933.4 eV, which corresponded to Cu^0^, and the binding energy of the Fe 2p peak was 706.97 eV and 711.72 eV, which corresponded to Fe^0^ and Fe^3+^, respectively [35]. In summary, we concluded that the compound existed in the form of Cu0, Fe0 and Fe^3+^. XPS narrow-spectrum scans of C(1s), O(1s), Fe(2p) and Cu(2p) before and after elimination were conducted as shown in Figure 14. The amount of change in the substances before and after the reaction was inferred from the peak area size. It was found that the iron and copper content decreased and the carbon and oxygen content increased, leading to the conclusion that elemental iron and copper play an important role in the adsorption of carmine dyes.

### 3.2. Experimental Results

In this investigation, the BBD of the RSM was adopted to evaluate the process variables for the removal of carmine using rGO/Fe/Cu nanocompounds and the single-factor investigation of this material for the removal of binary dyes mixed with carmine and Congo red. The experimental and predicted data for the removal of carmine from simulated wastewater are presented in Table 3. Figure 15 presents the line graph of the experimental and predicted values of the RSM. To describe the relationship between the decontamination efficiency and independent parameters, a multivariate analysis was carried out, and the quadratic model was as follows:Y = 84.73 − 1.78A − 3.88B +5.01C − 0.53D + 0.90AB + 7.07AC + 0.095AD + 4.67BC − 1.42BD + 5.38 CD + 3.92A^2^ + 1.49B^2^ − 6.67C^2^ + 1.91D^2^(6)
where Y is the decontamination efficiency of carmine; A is the contact time; B represents the initial pH; C stands for the dosage and D represents the concentration.

Analysis of variance (ANOVA) is designed to test the significance of the difference between the means of two or more samples [36]. Based on Table 4, the F-value of the model was derived as 38.19779, and the *p*-value was less than 0.05, which indicated that the model was significant. The F-values represented the effect of the operating parameters on the adsorption of carmine by the rGO/Fe/Cu nanocomposites. The order of importance of the operational parameters was C > B > A > D. Therefore, the model was of excellent applicability because of its high F-value and low *p*-value yielding more reliable conclusions.

Figure 16 presented the relationship between the probability of normal distribution. The points of the residuals on the plot followed a straight line, which indicated that the model prediction was accurate. Figure 17 showed the comparison of the experimental carmine decontamination percentage with the predicted values obtained from the model, and the value of the determination coefficient (Adj-R^2^ = 0.9409) is also presented in Figure 17. Excellent agreement was displayed between the experimental and predicted values of the carmine decontamination percentage. As shown in Figure 18, three-dimensional (3D) and contour (2D) response surface plots present the interaction between the two tested variables and the dye decontamination, while the other variables were kept at a fixed level. The experimental results indicated that the decontamination rate of carmine increased as the dosage of the material increased. This was because at high concentrations of the material, the ratio of the surface active sites to the dye molecules was high, and all molecules stuck to the surfaces of the materials. The increase in the number of effective collisions accelerated the reaction rate.

### 3.3. BP-ANN Model

The experimental data of the BP-ANN model was collected from the BBD in the response surface, and among the 29 sets of data, the first 24 sets were used for training, and the last 5 sets were experimental values (Table 5). The R^2^ value of the BP-ANN model was 0.9975, indicating that the predicted value of the BP neural network prediction model had a small difference from the expected value error and the correlation between the two was strong (Figure 19). The trained network performed accurately. As shown in Figure 20, the fourth neuron had the smallest MSE, from which it was inferred that the hidden layer contained four neurons (Figure 21). The data in Table 6 were obtained by GA optimization and were utilized as weights for the calculation of the Garson formula. The influence for each input variable on the output variable was calculated by the Garson equation using the weights. Table 7 indicates that the contribution of the dosage to dye decontamination was the highest (58.35%), followed by pH (18.96%), contact time (14.44%) and 8.23% for the concentration. Figure 22 illustrates the significance ranking of the factors fitted by R-studio, and Figure 23 shows the significance ranking of the factors obtained by R-gui. It can be seen that four factors’ weight analyses yielded consistent results, increasing the credibility of this conclusion. The R^2^ value of BP derived from the radial basis function was 0.98562 (Figure 24), which was slightly smaller than the value of ANN-GA, and the R^2^ value of the RBF was 0.90069, which indicated that the confidence level of the results obtained by this function and the reliability of the results were high.

### 3.4. Predicting the Carmine Decontamination Efficiency with BBD, ANN-PSO and ANN-GA

The maximum percentage of decontamination predicted by using the response surface BBD model was 95.8%, and the corresponding experimental value was 93.28% (Table 8) under the following conditions: an initial carmine concentration of 437.4 mg/L, pH of 5.09, dosage of 14.3 mg and contact time of 8.43 min. The performance of the ANN-PSO model indicated that the predicted decontamination efficiency was 96.13%, (Figure 25)and the corresponding experimental value was 95.79% under the following conditions: an initial carmine concentration of 764.3 mg/L, pH of 5.31, contact time of 12 min and dosage of 16.62 mg. The optimum values of the independent parameters for the ANN-GA were 581.3 mg/L for the initial carmine concentration, 5.85 for the initial pH, 14.49 mg for the dosage and 8.01 min for the contact time. The maximum decontamination efficiency predicted under this condition was 95.84%, (Figure 26)while the corresponding experimental value was 92.17%. For the above three, the direct absolute errors of the predicted and experimental values were 2.52, 0.34, and 3.67, respectively. It can be concluded that the ANN-PSO model was suitable for predicting the carmine decontamination by rGO/Fe/Cu nanohybrids. Therefore, the material had an excellent decontamination efficiency for carmine.

### 3.5. Adsorption Kinetics

The kinetic study of the adsorption process is mainly for describing the rate of solute adsorption by the adsorbent, and the kinetic model was used to fit the data to investigate the adsorption mechanism [37]. In this investigation, to study the mechanism of the rGO/Fe/Cu nanohybrids and the rate-controlling step of the process, adsorption experiments of the simulated dye wastewater with different initial concentrations were carried out at 180 W and pH = 5, and four kinetic models were used to evaluate and analyze the adsorption kinetics of the material.

The pseudo first-order adsorption model uses the Lagergren equation to calculate the adsorption rate [38]:(7)$$\frac{dq_{t}}{dt}=k_{1}\left(q_{e}-q_{t}\right)$$
where *q_e_* (mg/g) and *q_t_* (mg/g) are the amounts of carmine adsorbed at equilibrium and time *t* (min), respectively, and *K_1_* (min^−1^) is the equilibrium rate constant of quasi-primary adsorption. An integral of Equation (8) from *t* = 0 to *t* > 0 (*q* = 0 to *q* > 0) is obtained:(8)$$\ln\left(q_{e}-q_{t}\right)=lnqe-k_{1}t$$

The pseudo-secondary adsorption model is built on the basis that the rate control step is a chemical reaction or chemisorption through electron sharing or electron gain or loss with the pseudo-secondary kinetic equation, expressed as follows [39]:(9)$$\frac{dq_{t}}{dt}=k_{2}{\left(q_{e}-q_{t}\right)}^{2}$$
where *k*_2_ (g/mg/min) is the equilibrium rate constant for pseudo-secondary adsorption. Perform the integration of Equation (10) from *t* = 0 to *t* > 0 (*q* = 0 to *q* > 0) and in the form of a straight line as follows [40]:(10)$$\frac{t}{q_{t}}=\frac{1}{k_{2}q_{e}^{2}}+\frac{t}{q_{e}}$$

The intraparticle diffusion model was first proposed by Weber et al., and its expression is as follows [41]:qt=k3t12
where *k*_3_ is the intraparticle diffusion rate constant and *q_t_* is the amount of carmine adsorbed at time *t*, where *t*^1/2^ is the square root of time.

Elovich concluded that the adsorption rate decreased exponentially with the increase in adsorption on the surface of the adsorbent, and its simplified mathematical expression is as follows [42]:(11)qt=(1βE)ln(αEβE)+(1βE)lnt
where *α_E_* (mg/g/min) is the initial adsorption rate constant and *β_E_* (g/mg) is the desorption rate constant, which is also related to the degree of surface coverage and activation energy of chemisorption.

The parameters obtained from these four kinetic models are presented in Table 9, and it can be seen from the R^2^ values that the pseudo-secondary model described the kinetic experiments more satisfactorily than the pseudo-secondary, intraparticle diffusion and Elvoich kinetics, which indicates the adsorption process was mainly controlled by chemical interactions (Figure 27).

### 3.6. Adsorption Thermodynamics Study

The Langmuir equation can be applied to both physisorption and chemisorption, assuming that the adsorption is a single molecular layer on a uniform surface and ignores the lateral interactions between the adsorbed molecules. The Langmuir equation can be expressed as [43,44],
(12)$$\frac{C_{e}}{q_{e}}=\frac{1}{q_{m}K_{L}}+\frac{C_{e}}{q_{m}}$$
(13)$$R_{L}=\frac{1}{1+K_{L}C_{O}}$$
(14)$$q_{e}=\frac{q_{m}k_{L}C_{E}}{1+k_{L}C_{E}}$$
where *c_o_* and *c_e_* are the initial and final carmine concentrations (mg/L), respectively; *q_e_* is the adsorption amount of carmine at equilibrium (mg/g); *q_m_* is the maximum adsorption capacity; *K_L_* is the Langmuir constant (L/mg); and *R_L_* is the separation factor. The parameter *R_L_* indicates the nature of the isotherm as follows (Table 10) [45].

The Freundlich equation was suitable for physical adsorption and chemical adsorption. It was assumed that the heat of adsorption on a non-uniform surface decreased logarithmically as the surface coverage increased. Its expression can be written as [46],
(15)$$lnq_{e}=lnk+\frac{1}{n}lnC_{e}$$
where *k* and *n* are Freundlich constants related to the adsorption capacity and intensity of adsorption, respectively.

The Temkin equation, which describes chemisorption, assumes that the heat of adsorption decreases linearly with increasing adsorption due to the interaction between the adsorbent and the adsorbent and that the binding energy of adsorption is uniformly distributed [47]. The expression is
(16)$$q_{e}=alnK_{T}+alnC_{e}$$
where *a* is related to the heat of adsorption and *K_T_* (L/g) is the binding constant responding to the maximum binding energy at equilibrium.

A linear fit for the Langmuir and Freundlich models is displayed in Figure 28, and Table 11 presents the R^2^ values for the three thermodynamic studies. The results revealed that Freundlich’s linear fit R^2^ value (0.9709) was higher than that of Langmuir and Temkin’s linear fit. Thus, the adsorption of carmine on nanohybrids could be better described by Freundlich adsorption lines. The maximum adsorption capacity calculated by Langmuir was 1848.96, which once again proved the superior adsorption performance of the material.

### 3.7. rGo/Fe/Cu Removal Dual Dye

Industrial wastewater is diverse and is a mixture we generally used (in order to simulate more vividly) as a mixture of carmine and Congo red to simulate and explore the adsorption effect of the material for the mixed dye. The effect of the rGo/Fe/Cu nanocompounds on the adsorption of the binary dyes (carmine and Congo red) was investigated by single-factor experiments while controlling the following variables: contact time (2–30 min), dosage (10–30 mg), concentration (300–900 mg/L) and pH (3–9) (Figure 29). It was concluded that under optimal conditions (dosage of 20 mg, pH of 4, concentration of 500 mg/L and reaction time of 14 min), the removal of binary dyes could reach 96.85%. The high efficiency of the material for dye adsorption was further demonstrated.

## 4. Conclusions

Dyes are very difficult to treat once they enter the water because of their complex molecular structure. The efficient treatment of wastewater containing dyes has become a major research hotspot in recent years. Preparation of adsorbents with excellent adsorption effects and mechanical properties from inexpensive and environmentally friendly raw materials is the way to go for dye adsorption. In this work, graphene oxide was synthesized through a modified Hummers method with graphite powder as the main material, and then the rGO/Fe/Cu composites were prepared by the co-precipitation method. This research demonstrated that the mesoporous rGO/Fe/Cu nanocomposite is a good decontaminant for both carmine and the binary dyes (carmine and Congo red) from simulated wastewater, and the adsorption efficiency can reach a high level in a short period (<12 min). The material was characterized by XRD, SEM, EDS, Raman spectroscopy, N_2_ adsorption and XPS. The post-experimental material was also characterized, and it was found that the material did not change much before or after the experiment, providing the possibility of reusing it to operate the repeated experiments; that is, the rGO/Fe/Cu nanohybrids can be regenerated and have application prospects as a useful adsorbent for water treatment. The decontamination performance of rGO/Fe/Cu for the treatment of a carmine dye solution and binary dye solutions (containing carmine and Congo red) was successfully predicted and optimized by the BBD, ANN-PSO and ANN-GA. The absolute errors between the predicted and experimental results were 2.52, 0.34, and 3.67, respectively. The existence of a high degree of agreement between the predicted results and experimental results indicated that the ANN-PSO could be used effectively for the evaluation and optimization of the effects of the independent variables on carmine and binary dye (carmine and Congo red) decontamination. In addition, the experimental data were well fitted to the pseudo-second-order kinetic model. The maximum adsorption capacity was calculated as 1848.96 mg/g based on the Langmuir isothermal adsorption model. In summary, this investigation developed a promising mesoporous material for the treatment of wastewater containing the unitary or binary dyes of high concentrations with a short contact time and high efficiency, and ANN technology could offer great potential for pollutant removal applications. There is a great possibility for the material to be applied practically in wastewater treatment systems.

## Figures and Tables

**Figure 1 materials-14-05268-f001:**
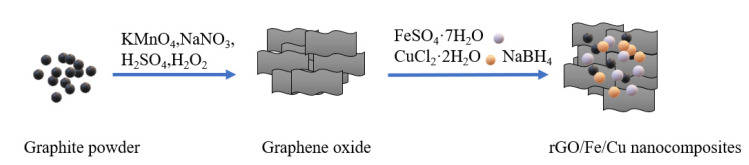
Schematic of rGO/Fe/Cu nanocomposite synthesis.

**Figure 2 materials-14-05268-f002:**
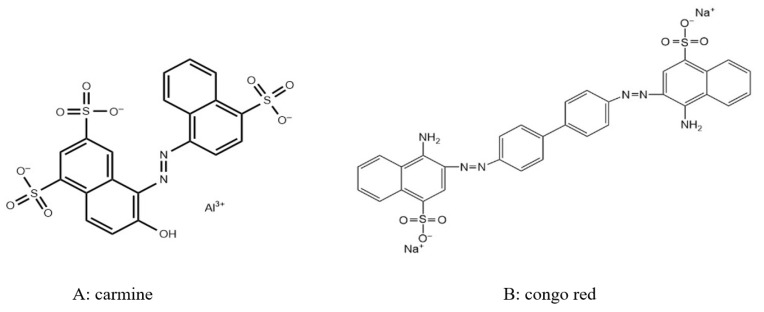
Structure of carmine (**A**) and Congo red (**B**).

**Figure 3 materials-14-05268-f003:**
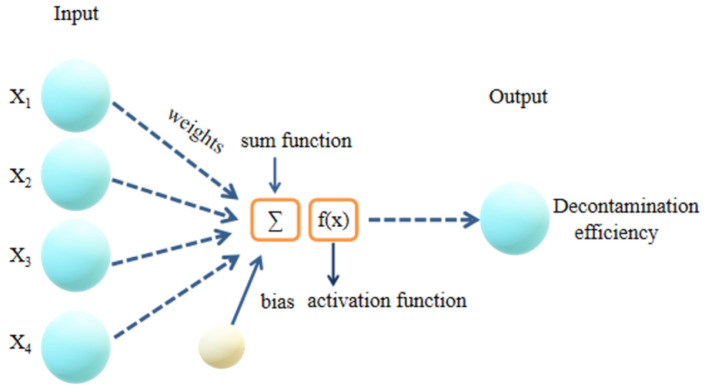
The schematic diagram for the artificial neuron model.

**Figure 4 materials-14-05268-f004:**
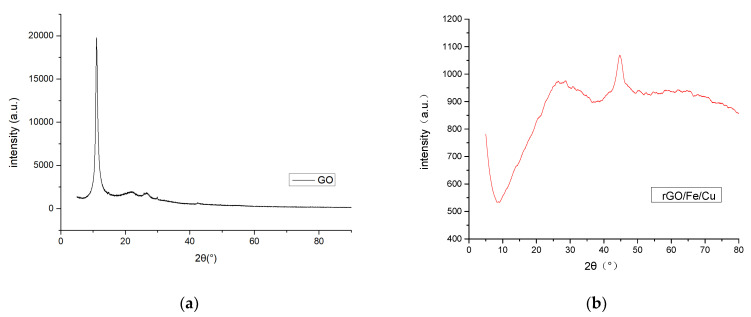
X-ray diffraction (XRD) patterns of (**a**) GO and (**b**) rGO/Fe/Cu nanocompounds.

**Figure 5 materials-14-05268-f005:**
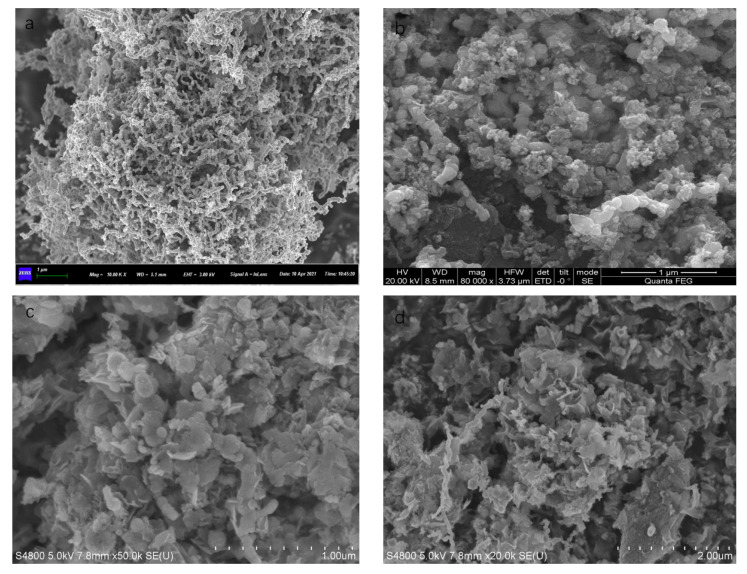
SEM images of Fe/Cu (**a**) and rGO/Fe/Cu (**b**) and the rGO/Fe/Cu at 1 um (**c**) and 2 um (**d**) after the experiment.

**Figure 6 materials-14-05268-f006:**
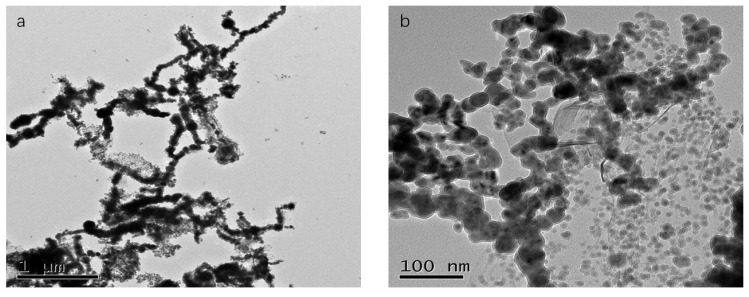
TEM images of Fe/Cu (**a**) and rGO/Fe/Cu (**b**).

**Figure 7 materials-14-05268-f007:**
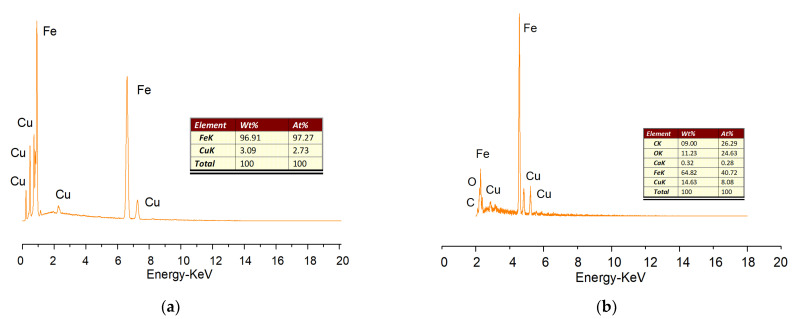
EDS spectra of the (**a**) Fe/Cu nanoparticles and (**b**) rGO/Fe/Cu nanocompounds.

**Figure 8 materials-14-05268-f008:**
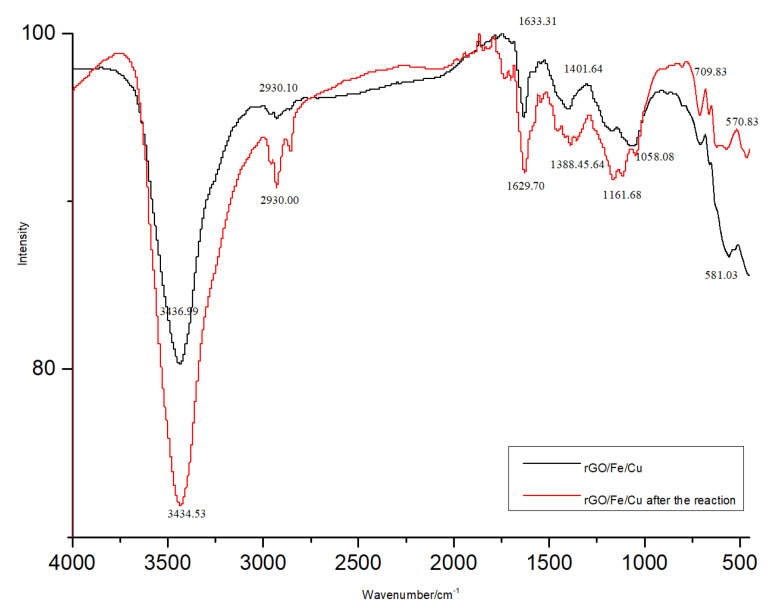
FTIR spectra of rGO/Fe/Cu nanocompounds before and after the reaction.

**Figure 9 materials-14-05268-f009:**
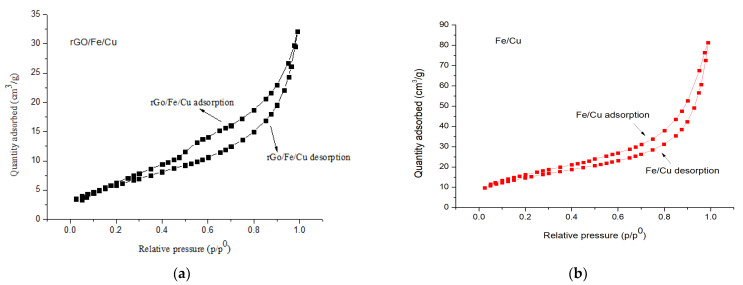
The adsorption and desorption isotherms of the (**a**) Fe/Cu nanoparticles and (**b**) rGO/Fe/Cu nanocompounds.

**Figure 10 materials-14-05268-f010:**
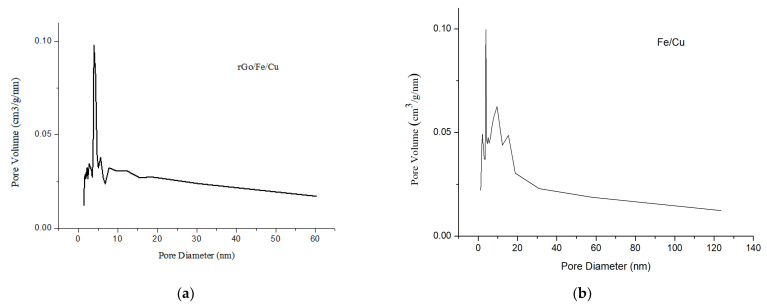
BJH pore size distribution curves of the (**a**) Fe/Cu nanoparticles and (**b**) rGO/Fe/Cu nanocompounds.

**Figure 11 materials-14-05268-f011:**
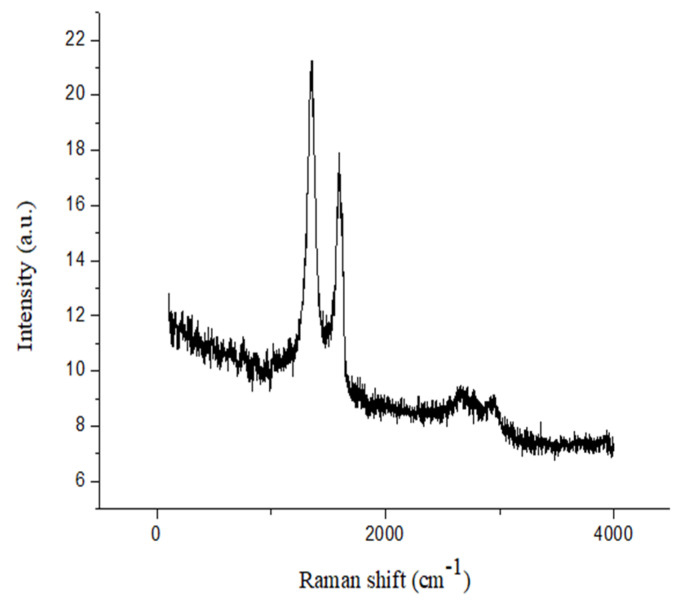
Raman spectra of rGO/Fe/Cu nanocompounds.

**Figure 12 materials-14-05268-f012:**
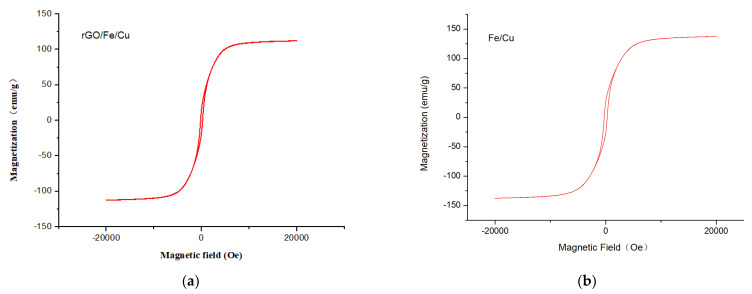
Magnetization hysteresis loop of the (**a**) Fe/Cu nanoparticles and (**b**) rGO/Fe/Cu nanocompounds at room temperature.

**Figure 13 materials-14-05268-f013:**
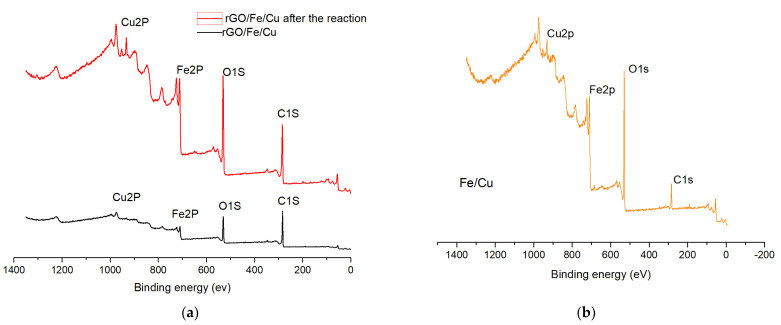
Long-range XPS spectra of (**a**) Fe/Cu nanoparticles and (**b**) rGO/Fe/Cu nanocompounds.

**Figure 14 materials-14-05268-f014:**
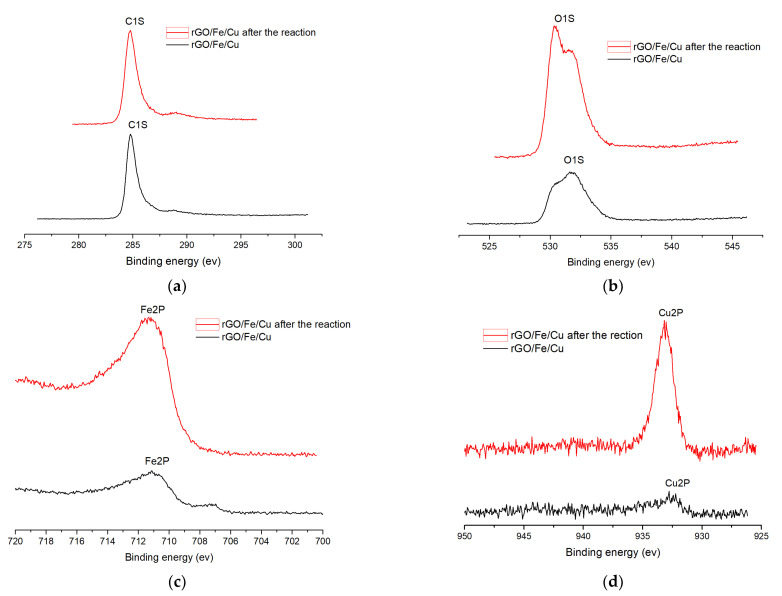
Photoelectron spectra of (**a**) C(1S); (**b**) O(1S); (**c**) Fe(2P) and (**d**) Cu(2P) before and after removing carmine by rGO/Fe/Cu.

**Figure 15 materials-14-05268-f015:**
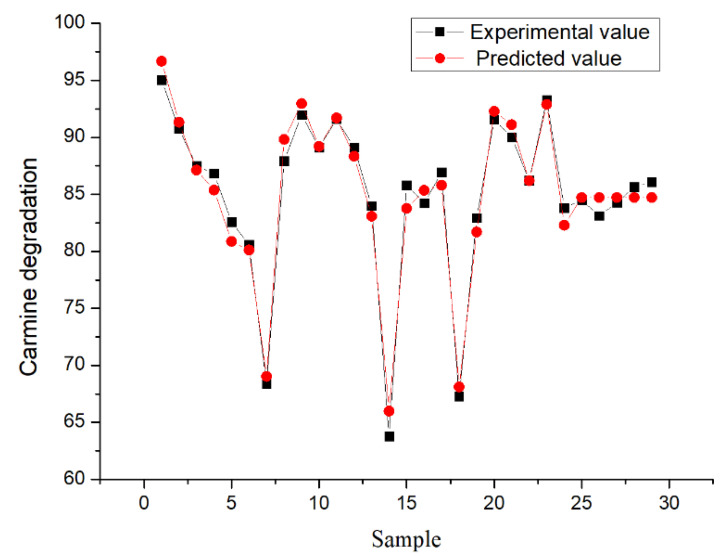
Comparison of experimental and predicted values for RSM.

**Figure 16 materials-14-05268-f016:**
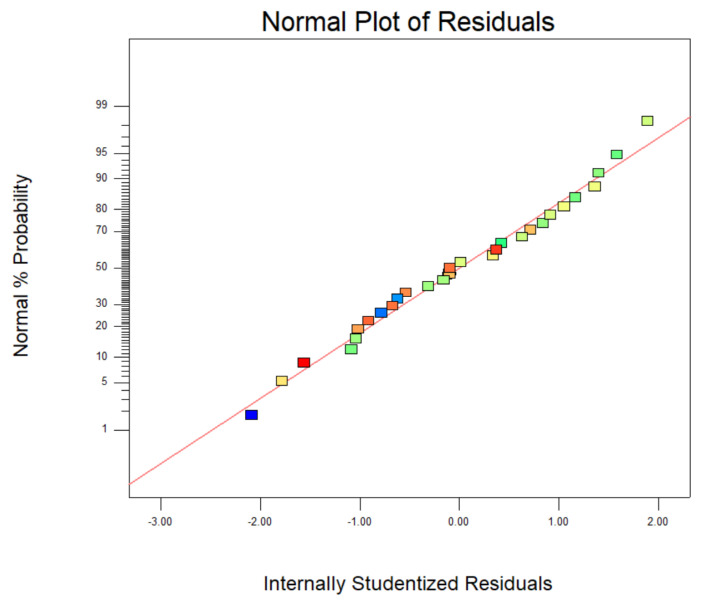
Normal probabilities and internal learning residuals.

**Figure 17 materials-14-05268-f017:**
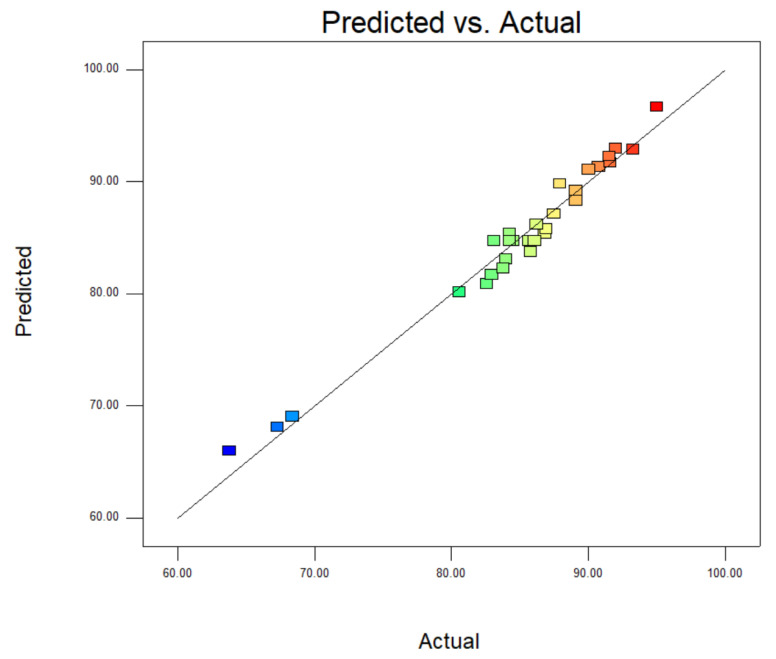
Comparison of predicted and actual values.

**Figure 18 materials-14-05268-f018:**
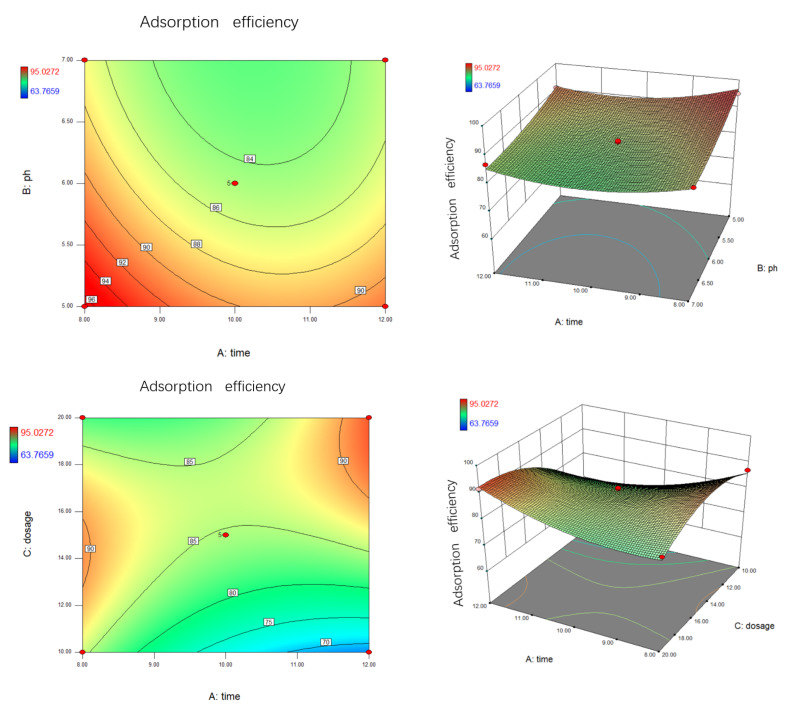
The 3D response surface and 2D contour line map plots for carmine decontamination.

**Figure 19 materials-14-05268-f019:**
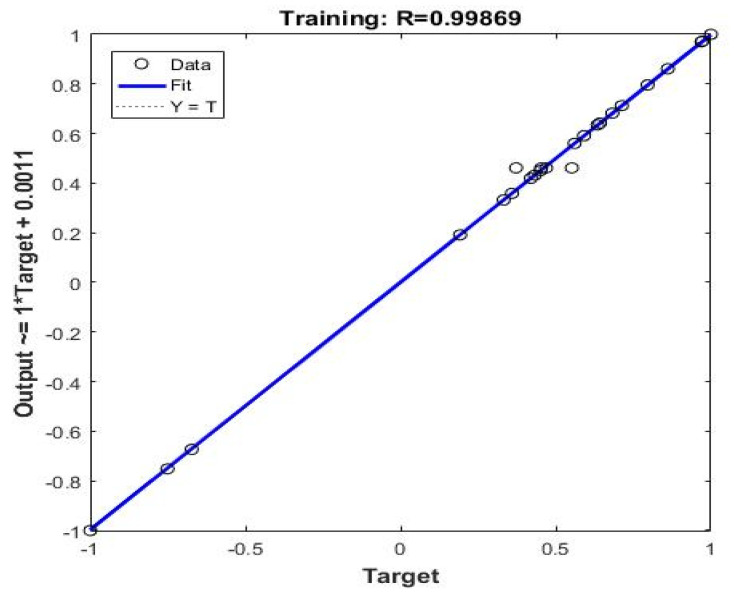
The experimental and predicted data of normalized decontamination.

**Figure 20 materials-14-05268-f020:**
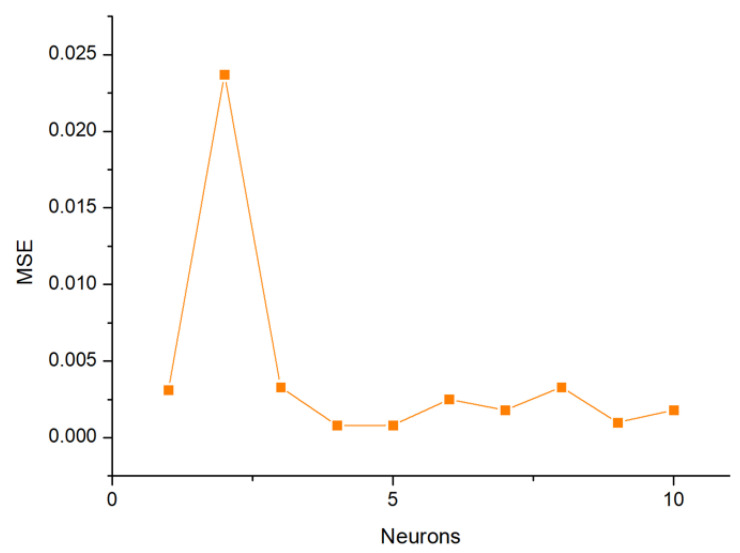
Mean square error (MSE) of the neurons in the BP-ANN model.

**Figure 21 materials-14-05268-f021:**
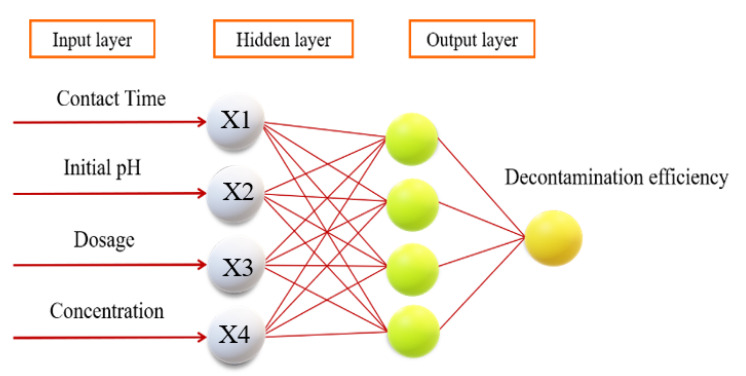
A backpropagation artificial neural network schematic diagram.

**Figure 22 materials-14-05268-f022:**
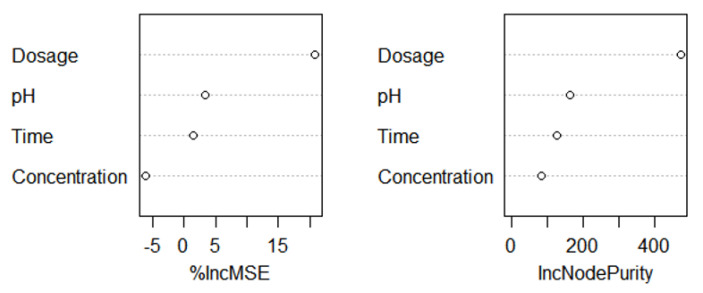
The relative impact of the input variables obtained by R-studio.

**Figure 23 materials-14-05268-f023:**
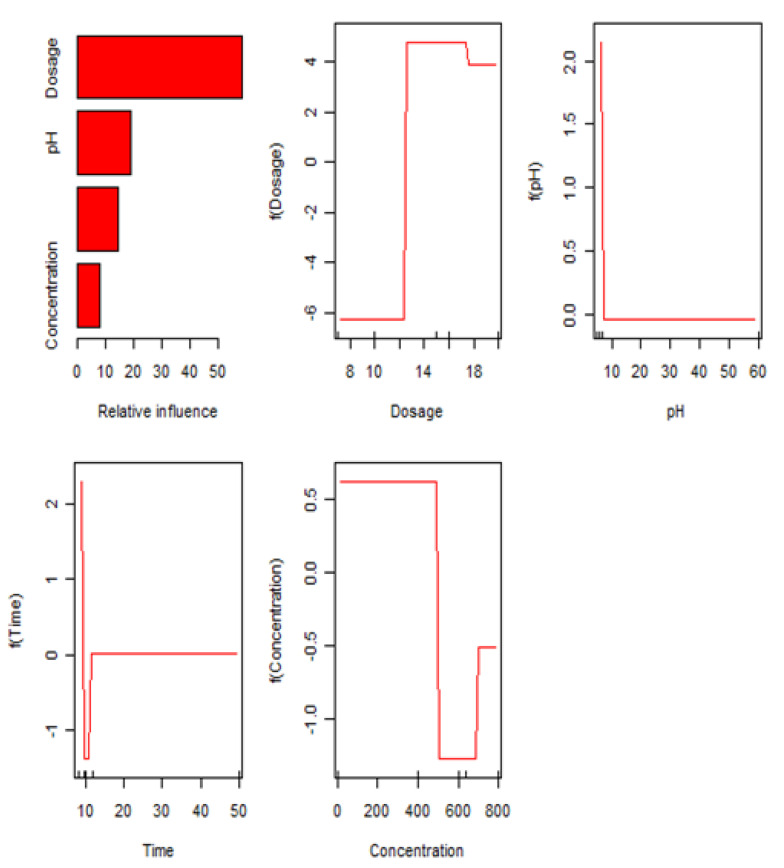
The relative impact of the input variables obtained by R-gui.

**Figure 24 materials-14-05268-f024:**
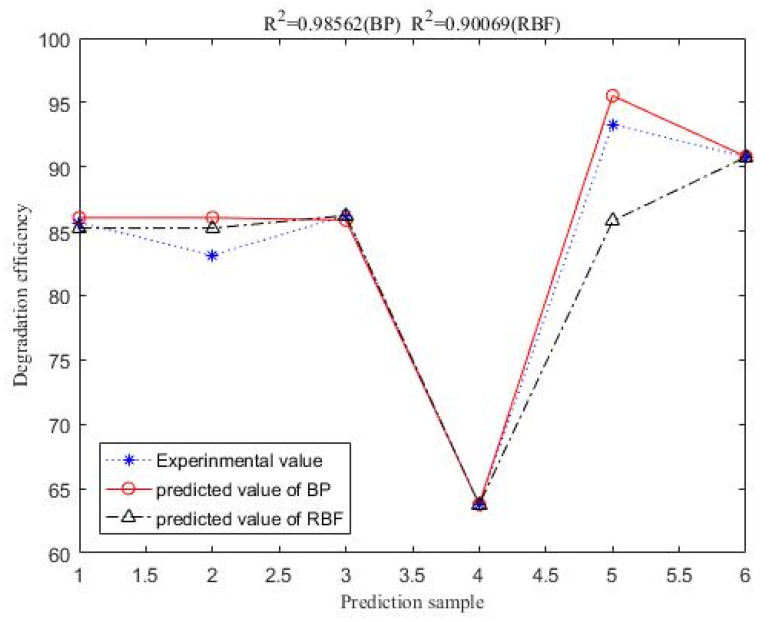
Comparison of the predicted value of the radial basis function and the predicted value of BP with the experimental value.

**Figure 25 materials-14-05268-f025:**
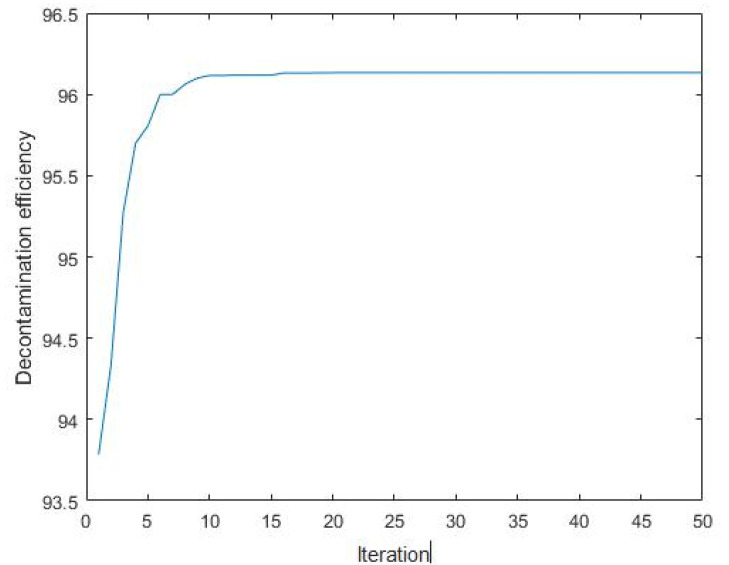
Decontamination efficiency versus iteration.

**Figure 26 materials-14-05268-f026:**
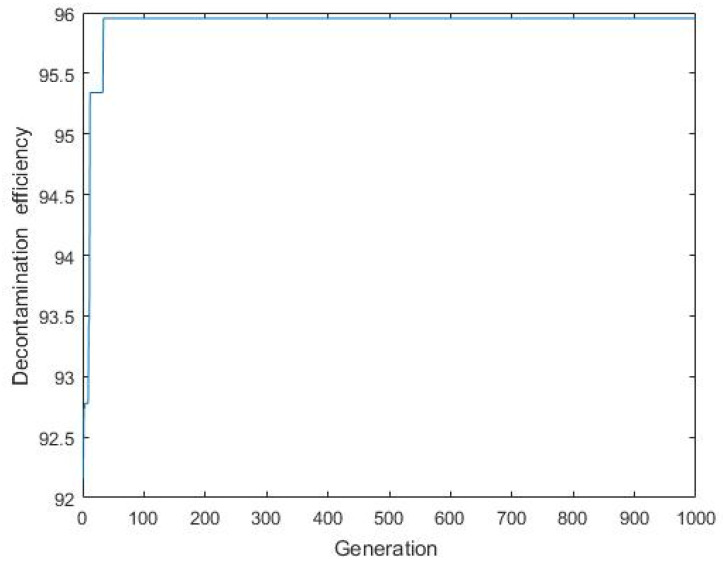
Decontamination efficiency versus generation.

**Figure 27 materials-14-05268-f027:**
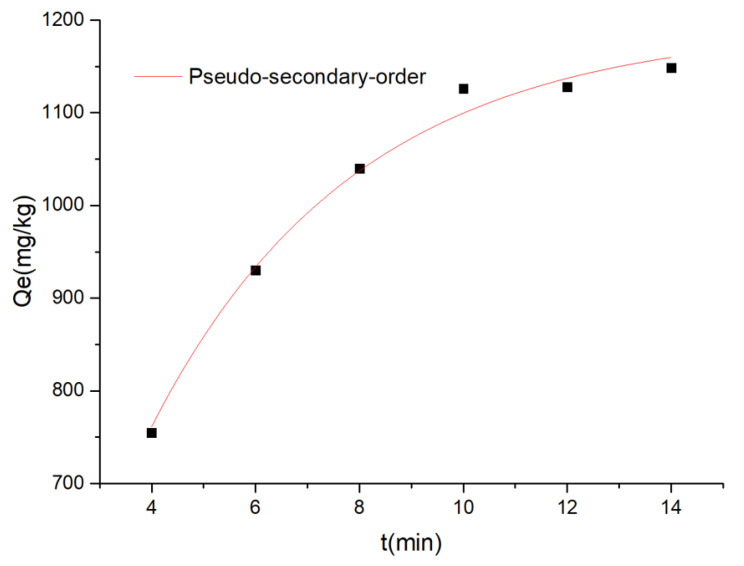
Study on the pseudo-secondary kinetics of carmine removal by rGO/Fe/Cu.

**Figure 28 materials-14-05268-f028:**
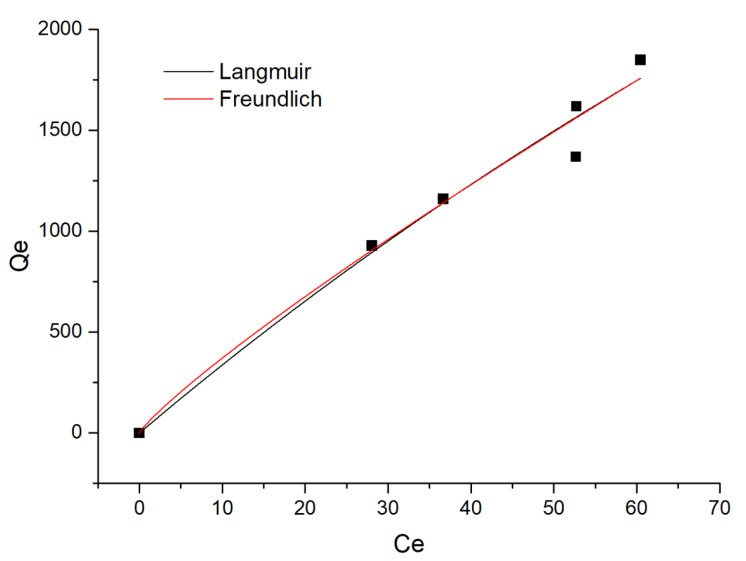
Thermodynamic study on the removal of carmine by rGO/Fe/Cu.

**Figure 29 materials-14-05268-f029:**
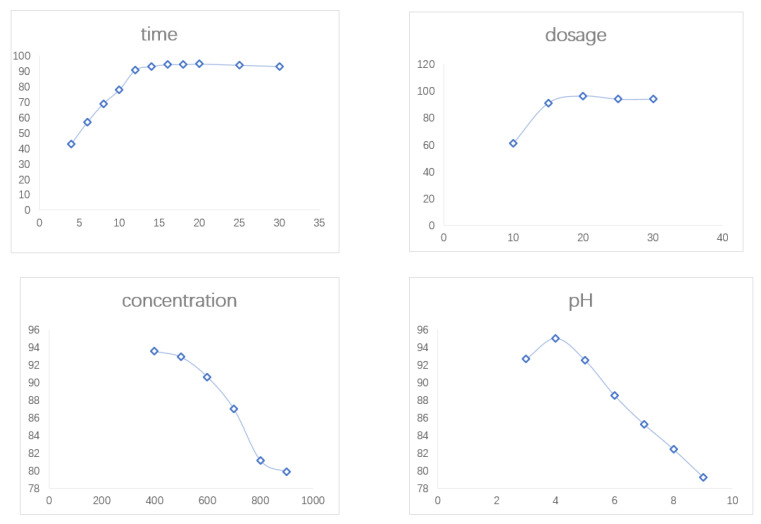
The single-factor experimental curve of binary dyes.

**Table 1 materials-14-05268-t001:** Chemical properties of carmine and Congo red.

Chemical Name	Carmine	Congo Red
Molecular formula	C_20_H_11_O_10_N_2_S_3_Na_3_	C_32_H_22_N_6_Na_2_O_6_S_2_
Molecular weight	604	696.68
Maximum wavelength λ	509 nm	497 nm
Solubility in water at 22 °C	≥10 g/100 mg	≥0.995 g/mL

**Table 2 materials-14-05268-t002:** Independent variables and levels in the experimental design.

Independent Variables	Unit	Code	Coded Variable Levels		
			−1	0	1
Contact time	min	A	8	10	12
pH	-	B	5	6	7
Dosage	mg	C	10	15	20
Concentration	mg/L	D	400	600	800

**Table 3 materials-14-05268-t003:** Comparison of the decontamination efficiency predicted by the BBD model with the experimental values.

Run	A (min)	B	C (mg)	D (mg/L)	Actual Value (%)	Predicted Value (%)	Absolute Error (%)
1	10	5	10	600	90.76	91.34	0.58
2	10	7	15	800	87.50	87.14	0.36
3	10	7	20	600	86.84	85.38	1.46
4	10	5	20	600	82.57	80.87	1.70
5	10	6	20	400	80.58	80.13	0.45
6	10	6	15	600	68.38	69.04	0.66
7	12	6	10	600	87.93	89.83	1.90
8	12	6	20	600	91.99	92.96	0.97
9	10	6	15	600	89.11	89.21	0.10
10	12	6	15	800	91.61	91.71	0.10
11	10	6	10	400	89.11	88.34	0.77
12	10	7	15	400	83.98	83.08	0.90
13	10	6	10	800	63.77	65.99	2.22
14	10	6	20	800	85.79	83.77	2.02
15	10	6	15	600	84.24	85.35	1.11
16	8	6	15	400	86.94	85.81	1.13
17	8	6	20	600	67.28	68.11	0.83
18	10	6	15	600	82.94	81.70	1.24
19	8	7	15	600	91.56	92.28	0.72
20	10	7	10	600	90.03	91.12	1.09
21	8	6	10	600	86.22	86.21	0.01
22	12	7	15	600	93.30	92.90	0.40
23	10	5	15	400	83.80	82.30	1.50
24	8	6	15	800	84.50	84.73	0.23
25	10	6	15	600	83.13	84.73	1.60
26	10	5	15	800	84.27	84.73	0.46
27	12	5	15	600	85.67	84.73	0.94
28	8	5	15	600	86.08	84.73	1.35
29	12	6	15	400	90.76	91.34	0.58
Mean absolute error (%)			0.87				

**Table 4 materials-14-05268-t004:** Analysis of variance (ANOVA) for the response surface quadratic model.

Source	Sum of Squares	Degree of Freedom	Mean Square	F-Value	*p*-Value	Significant or Not Significant
Model	1462.943	14	104.4959	38.19779	<0.0001	Significant
A (Time)	38.00306	1	38.00306	13.89177	0.0023	
B (pH)	180.3591	1	180.3591	65.9291	<0.0001	
C (Dosage)	301.4176	1	301.4176	110.1812	<0.0001	
D (Concentration)	3.381933	1	3.381933	1.236243	0.2849	
AB	3.228349	1	3.228349	1.180102	0.2957	
AC	199.9882	1	199.9882	73.10439	<0.0001	
AD	0.036267	1	0.036267	0.013257	0.9100	
BC	87.15988	1	87.15988	31.86073	<0.0001	
BD	8.11368	1	8.11368	2.965903	0.1070	
CD	115.9419	1	115.9419	42.38182	<0.0001	
A^2^	99.51958	1	99.51958	36.37873	<0.0001	
B^2^	14.43986	1	14.43986	5.278395	0.0375	
C^2^	288.5931	1	288.5931	105.4933	<0.0001	
D^2^	23.70787	1	23.70787	8.666257	0.0107	
Residual	38.29914	14	2.735653			
Lack of Fit	32.74903	10	3.274903	2.360244	0.2116	Not Significant
Pure Error	5.550108	4	1.387527			
Cor Total	1501.242	28				

**Table 5 materials-14-05268-t005:** The experimental design matrix of the backpropagation (BP) artificial neural network (ANN) model.

Runs	*X*_1_ (min)	*X* _2_	*X*_3_ (mg)	*X*_4_ (mg/L)	Experimental Value (%)	Predicted Value (%)
1	10	5	10	600	83.9843	81.9819
2	10	7	15	800	83.7990	85.0323
3	10	7	20	600	84.2445	83.4627
4	10	5	20	600	85.7910	88.5915
5	10	6	20	400	80.5823	86.7515
6	10	6	15	600	84.2665	85.1908
7	12	6	10	600	67.2769	83.0775
8	12	6	20	600	91.5635	84.6104
9	10	6	15	600	83.1270	85.3418
10	12	6	15	800	89.1085	85.7064
11	10	6	10	400	82.5666	80.6683
12	10	7	15	400	86.2249	85.1616
13	10	6	10	800	68.3817	82.6676
14	10	6	20	800	87.9326	85.2220
15	10	6	15	600	85.6656	85.0334
16	8	6	15	400	91.9886	87.1420
17	8	6	20	600	82.9441	88.2917
18	10	6	15	600	84.4953	84.4836
19	8	7	15	600	87.5047	85.6040
20	10	7	10	600	63.7659	80.8439
21	8	6	10	600	86.9410	80.3149
22	12	7	15	600	86.8359	84.3499
23	10	5	15	400	90.0294	88.1946
24	8	6	15	800	91.6078	86.8738
25	10	6	15	600	86.0846	85.5328
* 26	10	5	15	800	93.3004	87.2787
* 27	12	5	15	600	90.7649	87.8442
* 28	8	5	15	600	95.0272	88.0859
* 29	12	6	15	400	89.1085	86.5487

* stands for test set.

**Table 6 materials-14-05268-t006:** Weights and biases of the BP-ANN in the input hidden layers (wi and bi) and the hidden output layers (wj and bj).

Number of Neurons	w_i_				Input Bias	Layer Weights	Layer Bias
	Input Weights						
	Contact Time	Initial pH	Dosage	Concentration			
1	1.4272	−1.4802	0.9073	2.4895	2.4895	0.7555	−0.1334
2	−1.7844	−1.6223	0.1750	−1.9363	−1.9363	0.7555	
3	1.4870	1.4928	0.5855	−1.3831	−1.3831	0.7555	
4	−1.1910	0.5800	1.5085	0.8298	0.8298	0.7555	
5	0.6114	−0.9833	−1.6289	−0.2766	−0.2766	0.7555	
6	0.4390	−0.4901	−0.9972	0.2766	0.2766	0.7555	
7	0.6983	−1.5074	−1.8098	0.8298	0.8298	0.7555	
8	−0.5684	−0.9470	−1.2455	−1.3831	−1.3831	0.7555	
9	0.7825	−1.5754	1.4324	−1.9363	−1.9363	0.7555	
10	1.6152	−0.8738	1.1511	−2.4895	−2.4895	0.7555	

**Table 7 materials-14-05268-t007:** The relative influence of the input variables.

Input Variables	Relative Significance (%)	Order
Contact Time	14.44%	3
Initial pH	18.96%	2
Dosage	58.35%	1
Concentration	8.23%	4

**Table 8 materials-14-05268-t008:** Comparison of the predicted percentage of decontamination of carmine by the BBD, ANN-PSO and ANN-GA models with the experimental results.

Models	Independent Parameters				Prediction (%)	Experiment (%)	Absolute Error (%)
	A	B	C	D			
BBD	8.43	5.09	14.30	437.4	95.80	93.28	2.52
ANN-PSO	12.00	5.31	16.62	764.3	96.13	95.79	0.34
ANN-GA	8.01	5.85	14.49	581.3	95.84	92.17	3.67

**Table 9 materials-14-05268-t009:** The R^2^ values of the kinetics for carmine adsorption on rGO/Fe/Cu nanohybrids.

Kinetic Models	Values of R^2^
Pseudo-first-order	0.9897
Pseudo-second-order	0.9931
Intraparticle diffusion	0.9044
Elovich	0.9496

**Table 10 materials-14-05268-t010:** Rules for judging the type of adsorption isotherm by *R_L_* value.

Values of R_L_	Type of Adsorption Nature
*R_L_* > 1	Unfavorable
*R_L_* = 1	Linear
0 < *R_L_* < 1	Favorable
*R_L_* = 0	Irreversible

**Table 11 materials-14-05268-t011:** The R^2^ values of the thermodynamics for carmine adsorption onto rGO/Fe/Cu nanohybrids.

Thermodynamics Models	Values of R^2^
Langmuir	0.8704
Freundlich	0.9709
Temkin	0.8837

## Data Availability

The data that support the findings of this study are available from the corresponding author upon reasonable request.

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
