# Peer review of "Binary Dye Removal from Simulated Wastewater Using Reduced Graphene Oxide Loaded with Fe-Cu Bimetallic Nanocomposites Combined with an Artificial Neural Network"

_materials, 2021, doi:10.3390/ma14185268_

Round 1
Reviewer 1 Report
This manuscript presented the main results obtained regarding the use of the mesoporous rGO/Fe/Cu nanocomposite as decontaminant for both carmine and the binary dyes of carmine and congo red in simulated wastewater, and the adsorption efficiency determination.
The manuscript is original, interesting and well-organized.
The English language used in the article should be revised.
At the Introduction, I think it must be added the following reference: [Usman Farooq et al., Efficient transformation in characteristics of cations supported-reduced graphene oxide nanocomposites for the destruction of trichloroethane, Applied Catalysis A, General 544 (2017) 10–20]
Verify molecular formula of carmine (page 5).
At preparation of GO, please mention at the step of H2O2 adding, the temperature and reaction time.
Also, the GO preparation seems to be similar with the GO preparation from reference [27] and [29], so mention these references in text.
Please make the following corrections:
Page 1, line 22 “three times” instead of “there times”
Page 5, line 177, “carmine and congo red” instead of “methylene blue”
Page 6, line 202, reformulate the sentence something like “The diffractograms of the prepared Fe/Cu nanoparticles….”
Page 6, line 204, “source” instead of “target”
Page 7, line 254, Figure 3 – missing reference
Page 8, line 275, Reference [29] is wrong correlated because in your reference list, [29] is Ruan et al.; reformulate, because in RGO diffraction, a smooth peak appears at 24-25°
Author Response
- This manuscript presented the main results obtained regarding the use of the mesoporous rGO/Fe/Cu nanocomposite as decontaminant for both carmine and the binary dyes of carmine and congo red in simulated wastewater, and the adsorption efficiency determination. The manuscript is original, interesting and well-organized. The English language used in the article should be revised.
Responses: Thank you very much for your valuable comments and suggestions. The language of our manuscript has been polished.
- At the Introduction, I think it must be added the following reference: [Usman Farooq et al., Efficient transformation in characteristics of cations supported-reduced graphene oxide nanocomposites for the destruction of trichloroethane, Applied Catalysis A, General 544 (2017) 10–20].
Responses: Thank you very much for your valuable comments and suggestions. We have changed the text according to your comments.
- Verify molecular formula of carmine (page 5).
Responses: Thank you very much for your valuable comments and suggestions. We have verified the molecular formula of carmine.
- At preparation of GO, please mention at the step of H2O2 adding, the temperature and reaction time.
Responses: Thank you very much for your valuable comments and suggestions. We have changed the text according to your comments.
- Also, the GO preparation seems to be similar with the GO preparation from reference [27] and [29], so mention these references in text.
Responses: Thank you very much for your valuable comments and suggestions. We have added literature citations to the preparation process.
- Please make the following corrections:
Page 1, line 22 “three times” instead of “there times”.
Responses: Thank you very much for your valuable comments and suggestions. We have changed the text according to your comments.
Page 5, line 177, “carmine and congo red” instead of “methylene blue”.
Responses: Thank you very much for your valuable comments and suggestions. We have changed the text according to your comments.
Page 6, line 202, reformulate the sentence something like “The diffractograms of the prepared Fe/Cu nanoparticles….”.
Responses: Thank you very much for your valuable comments and suggestions. We have changed the text according to your comments.
Page 6, line 204, “source” instead of “target”.
Responses: Thank you very much for your valuable comments and suggestions. We have changed the text according to your comments.
Page 7, line 254, Figure 3 – missing reference.
Responses: Thank you very much for your valuable comments and suggestions. We have changed the text according to your comments.
Page 8, line 275, Reference [29] is wrong correlated because in your reference list, [29] is Ruan et al.; reformulate, because in RGO diffraction, a smooth peak appears at 24-25.
Responses: Thank you very much for your valuable comments and suggestions. We have changed the text according to your comments.
Reviewer 2 Report
The abstract is adequate and does not need any major revisions.
The section on artificial intelligence in the introduction lumps optimization methods with function approximation methods and does not do enough to distinguish between the difference.
The methodology section on experimental methods is adequate. However, for the ANN-optimization methodology should have mentioned more on the data available, split between training, validation, and testing data. Also, no mention was made of how the features for the neural network were selected.
In the results section, the authors could have used more performance metrics for the various approaches. In terms of presentation of the results, more data points could have been used to avoid the non-smooth appearance in the graphs produced. Figures 20 are examples of the "jagged" appearance due to a lack of enough data points. Figures 27 and 28 have too much information in the boxes and are not neccessary.
The last figure is mislabelled and should be Figure 29 and not 21. Also, the background with the grid should be removed.
Author Response
Response to Reviewer 2 Comments:
- The abstract is adequate and does not need any major revisions.
Responses: Thank you very much for your valuable comments and suggestions. We have made a minor modification.
- The section on artificial intelligence in the introduction lumps optimization methods with function approximation methods and does not do enough to distinguish between the difference.
Responses: Thank you very much for your valuable comments and suggestions. We have done some distinctions between artificial intelligence lumps optimization methods and function approximation methods.
- The methodology section on experimental methods is adequate. However, for the ANN-optimization methodology should have mentioned more on the data available, split between training, validation, and testing data. Also, no mention was made of how the features for the neural network were selected.
Responses: Thank you very much for your valuable comments and suggestions. We have added some expressions about ANN, and compare the difference between the experimental and predicted values of several algorithms and the R2 value to calculate the better one, and then improve the efficiency in the next process.
- In the results section, the authors could have used more performance metrics for the various approaches. In terms of presentation of the results, more data points could have been used to avoid the non-smooth appearance in the graphs produced. Figures 20 are examples of the "jagged" appearance due to a lack of enough data points. Figures 27 and 28 have too much information in the boxes and are not neccessary.
Responses: Thank you very much for your valuable comments and suggestions. We have changed the text according to your comments.
- The last figure is mislabelled and should be Figure 29 and not 21. Also, the background with the grid should be removed.
Responses: Thank you very much for your valuable comments and suggestions. We have changed the text according to your comments.
Reviewer 3 Report
The manuscript describes the synthesis of reduced graphene oxide supported Fe-Cu bimetallic nanocomposites for dye removal. In addition, the synthesized materials were operated under optimized parameter through artificial intelligent. This research targets important classes of dye recovery and remediation. Herein, there are a number of improvements and areas of concern which must be addressed before publication.
- Line 272 to 282 presents the XRD result for characterizing rGO/Fe/Cu nanocomposite. The diffraction angles are corresponding to specific facets of materials. However, the XRD result seems inconsistency. The main peak located at 44.6o assigning as nZVI. But the remaining peak at 35.5 and 65o are not shown in Figure obviously. Could authors explain them more in detail. Maybe it is better to confirm with XRD database, like JCPDS.
- The adsorption behavior and performance would be determined by surface area. But this research just show the pore size through BET analysis. What is the surface area of rGO/Fe/Cu and Fe/Cu?
- The adsorption behavior and performance is not only determined by adsorption environment but also by materials properties. However, the optimized parameters through ANN just considered the environment issue. How about the material composition, like the ratio of Fe/Cu, ID/IG etc.
- From XPS spectrum, the intensity of Fe/Cu was reduced after adsorption. Therefore, the removal mechanism should be clarified and be discussed. The results demonstrated that the adsorption look like not the only way for dye removal. Authors should discussed this observation in detail.
- Figure 15 presents the degradation of dye for rGO/Fe/Cu. Please clarify the main mechanism for dye removal.
- In the dual dye adsorption section, these two dyes (Carmine and Congo red) perform negative charge. Does the competitive or selective adsorption occur in this study?
- The figure caption should be corrected and remarked. Figure 5A and 5a, 5B and 5b, and C and 5c, should be better to name as 5a to 5f.
Author Response
Response to Reviewer 3 Comments:
- Line 272 to 282 presents the XRD result for characterizing rGO/Fe/Cu nanocomposite. The diffraction angles are corresponding to specific facets of materials. However, the XRD result seems inconsistency. The main peak located at 44.6o assigning as nZVI. But the remaining peak at 35.5 and 65o are not shown in Figure obviously. Could authors explain them more in detail. Maybe it is better to confirm with XRD database, like JCPDS.
Responses: Thank you very much for your valuable comments and suggestions. We have changed the text according to your comments.
- The adsorption behavior and performance would be determined by surface area. But this research just show the pore size through BET analysis. What is the surface area of rGO/Fe/Cu and Fe/Cu?
Responses: Thank you very much for your valuable comments and suggestions. We have changed the text according to your comments.
- The adsorption behavior and performance is not only determined by adsorption environment but also by materials properties. However, the optimized parameters through ANN just considered the environment issue. How about the material composition, like the ratio of Fe/Cu, ID/IG etc.
Responses: Thank you very much for your valuable comments and suggestions. The ID/IG intensity (1.18) of rGO/Fe/Cu nanohybrids was greater than 1, indicating a large number of structural defects in rGO/Fe/Cu nanohybrids. The prepared graphene oxide loaded with Fe/Cu in a ratio of 5:1 showed better adsorption compared with the graphene oxide loaded with zero-valent iron nanoparticles made by Ruan et al., which indicates a synergistic effect of Fe/Cu.
- From XPS spectrum, the intensity of Fe/Cu was reduced after adsorption. Therefore, the removal mechanism should be clarified and be discussed. The results demonstrated that the adsorption look like not the only way for dye removal. Authors should discussed this observation in detail.
Responses: Thank you very much for your valuable comments and suggestions. It is also known that zero-valent iron nanoparticles are good catalysts, since they are present before and after the reaction, so it can be concluded that the material is not only an adsorbent but also can be considered as a catalyst.
- Figure 15 presents the degradation of dye for rGO/Fe/Cu. Please clarify the main mechanism for dye removal.
Responses: Thank you very much for your valuable comments and suggestions. Figure 15 presented the line graph of the experimental and predicted values of RSM. The changes of the materials, as well as the functional groups before and after the reaction, were found to be more obvious before and after the reaction of hydroxyl and Fe/Cu elements, so it was deduced that they were the main adsorption groups.
- In the dual dye adsorption section, these two dyes (Carmine and Congo red) perform negative charge. Does the competitive or selective adsorption occur in this study?
Responses: Thank you very much for your valuable comments and suggestions. To verify whether the material is competitive or selective adsorption, three sets of parallel experiments were added. rGO/Fe/Cu nanohybrids were first used to adsorb carmine dyes, and after complete reaction, congo red dyes were added and sonicated for several minutes, and it was found that none of them could continue to adsorb, so it was inferred that the material was competitively adsorbed.
- The figure caption should be corrected and remarked. Figure 5A and 5a, 5B and 5b, and C and 5c, should be better to name as 5a to 5f.
Responses: Thank you very much for your valuable comments and suggestions. We have changed the text according to your comments.
Round 2
Reviewer 1 Report
Given the novelty of the work and the fact that the authors made the suggested corrections, I recommend to accept this article for publication in its present form.